# Reconstruction of Partial Hypopharyngeal Defects following Total Laryngectomy: A Systematic Review and Meta-Analysis

**DOI:** 10.3390/cancers16101804

**Published:** 2024-05-08

**Authors:** Anthony M. Tonsbeek, Roxy Leidelmeijer, Caroline A. Hundepool, Liron S. Duraku, Mark J. W. Van der Oest, Aniel Sewnaik, Marc A. M. Mureau

**Affiliations:** 1Department of Plastic & Reconstructive Surgery, Erasmus MC Cancer Institute, University Medical Center Rotterdam, 3015 GD Rotterdam, The Netherlandsm.mureau@erasmusmc.nl (M.A.M.M.); 2Department of Plastic, Reconstructive and Hand Surgery, Amsterdam UMC, 1105 AZ Amsterdam, The Netherlands; 3Department of Otorhinolaryngology and Head and Neck Surgery, Erasmus MC Cancer Institute, University Medical Center Rotterdam, 3015 GD Rotterdam, The Netherlands

**Keywords:** head and neck cancer, reconstruction, fasciocutaneous, pectoralis major flap, anterolateral thigh flap, radial forearm flap, postoperative complications, functional outcomes

## Abstract

**Simple Summary:**

Although several techniques exist for reconstructing partial hypopharyngeal defects following a total laryngectomy, no international consensus has been reached to date. As a result, there currently are large differences between institutions with regard to the flap types used to perform these reconstructions. The aim of this systematic review and meta-analysis was to examine the complication rates and functional results of commonly used reconstructive techniques for hypopharyngeal defects. Pectoralis major myofascial flaps showed promising results compared to free-flap reconstructions, with similar rates of fistulas, strictures and flap failure. In contrast, pectoralis major myocutaneous flaps had a significantly higher fistula rate (34%) in comparison to other flap types (range: 7–17%), whereas no differences were observed for strictures, flap failure or oral intake. Free flaps and pectoralis major myofascial flaps should be considered the preferred methods for the reconstruction of partial hypopharyngeal defects following total laryngectomy.

**Abstract:**

Background: Various operative techniques exist to reconstruct partial hypopharyngeal defects following total laryngectomy. The current study aimed to investigate and compare complications and functional results following commonly used reconstructive techniques. Methods: A systematic review and meta-analysis were performed using studies that investigated outcomes after the reconstruction of a partial hypopharyngeal defect. The outcomes of interest were fistulas, strictures, flap failure, swallowing function and postoperative speech. Results: Of the 4035 studies identified, 23 were included in this review. Four common reconstructive techniques were reported, with a total of 794 patients: (1) pectoralis major myocutaneous and (2) myofascial flap, (3) anterolateral thigh free flap and (4) radial forearm free flap. Fistulas occurred significantly more often than pectoralis major myocutaneous flaps (34%, 95% CI 23–47%) compared with other flaps (*p* < 0.001). No significant differences in the rates of strictures or flap failure were observed. Pectoralis major myofascial flaps were non-inferior to free-flap reconstructions. Insufficient data were available to assess speech results between flap types. Conclusion: Pectoralis myocutaneous flaps should not be the preferred method of reconstruction for most patients, considering their significantly higher rate of fistulas. In contrast, pectoralis major myofascial flaps yield promising results compared to free-flap reconstructions, warranting further investigation.

## 1. Introduction

The reconstruction of a partial hypopharyngeal defect following total laryngectomy (TLE) and subtotal pharyngectomy is a complex procedure, aiming to restore the continuity of the upper digestive tract and speech, if there is insufficient hypopharyngeal mucosa to achieve primary closure. A reconstruction is required if less than approximately 3 cm of stretched hypopharyngeal mucosa remains, in order to prevent a stricture of the neo-hypopharynx [1]. Because of the extensive nature of this surgical (salvage) procedure in a vulnerable group of patients, complications are frequently encountered. Postoperative complications in this setting are relatively common and may severely affect health-related quality of life in both the short and long term [2]. Consequently, creating a durable and functional neo-hypopharynx using a reconstructive technique that carries minimal procedure-related morbidity is crucial.

Over the years, various techniques have been described in the literature to reconstruct partial hypopharyngeal defects. The pedicled pectoralis major myocutaneous (PMMC) flap, first described by Ariyan in 1979, was historically used to reconstruct these defects as a reliable workhorse flap [3]. Using the remaining hypopharyngeal tissue and the skin island of the PMMC flap, a patch or inlay reconstruction restores the pharyngeal conduit, thereby creating a cutaneous inner-lining of the neo-hypopharynx. However, reports of inherent disadvantages of the PMMC flap in oncological head and neck reconstructions tempered the initial popularity and widespread use due to its bulkiness, poor pliability, relatively high rates of partial skin island necrosis, fistulas and associated donor-site morbidity [4,5,6,7]. Therefore, reconstructive surgeons pursued alternative techniques and, with the advent of reconstructive microsurgery, the radial forearm free flap (RFFF) and the anterolateral thigh free flap (ALTFF) gained popularity because both seemed to provide advantages over the traditional PMMC flap. Currently, some institutions advocate the use of free flaps over PMMC flaps in head and neck reconstruction, while other institutions still consider the latter to be the first choice of reconstruction. Nevertheless, few studies have investigated and compared the outcomes of regional pedicled flaps with free flaps, specifically in the context of a partial hypopharyngeal defect following TLE. Due to the controversy about which technique(s) should be considered primarily, many institutional differences and arbitrariness in treatment protocols exist. In the absence of objective evidence, it remains unknown which techniques lead to favorable functional results while minimizing the risk of complications.

Comparing the outcomes of commonly used reconstructive techniques is essential to limit procedure-related morbidity for patients, while optimizing functional results such as swallowing and speech. Therefore, the aim of this study was to investigate and compare postoperative complications and functional results following commonly used reconstructive techniques for partial hypopharyngeal defects after TLE.

## 2. Materials and Methods

A systematic literature review was performed to identify all studies investigating reconstructive methods for partial hypopharyngeal defects following TLE, using the Embase, Medline, Web of Science and Cochrane libraries. Studies reported in English between 1979 and 2023 were analyzed according to the PRISMA 2020 statement [8]. Articles were first assessed for eligibility based on title and abstract. The full-text article was obtained and evaluated if the study was not excluded based on the abstract. All abstracts and full-text articles were assessed separately by two independent reviewers (A.M.T. and R.L.). In addition, we added our own retrospective single-center data, including all consecutive patients who underwent a pectoralis major-flap reconstruction following TLE with a partial hypopharyngectomy between 1 January 2000 and 1 April 2022, and a consecutive cohort of patients that received an RFFF reconstruction [9].

The inclusion and exclusion criteria that were applied are shown in Figure 1. A flowchart of the inclusion process is shown in Figure 2. The complete search strategy can be found in the online supplement (Section A.1). All included studies were assessed on their quality and risk of bias using the National Institute of Health quality assessment tool, as shown in the online supplement (Table A1) [10].

The primary outcome measures of interest were the rates of fistulas and strictures. Since the majority of studies did not specify any details on fistula management, fistulas were generally defined as all anomalous connections between the hypopharynx and other anatomical structures, regardless of their management. If possible, separate categories of surgically treated and conservatively treated fistulas were made. Conservatively treated fistulas were defined as suture line leakages which were resolved spontaneously after conservative measures (e.g., prolonged tube feeding, antibiotics and/or anticholinergics). Moreover, while none of the studies that reported stricture rates mentioned a clear definition of strictures, most studies described a stricture as a functional stenosis requiring therapy, mainly using (balloon) dilation. ‘Malignant strictures’ due to local tumor recurrences were excluded.

The secondary outcome parameters included flap failure, speech and swallowing function. Studies on swallowing function and dietary intake showed many different classification methods. To categorize the results for the current review, the outcomes were divided into four groups: (1) satisfactory oral intake (solid or soft diet), (2) liquid diet, (3) any oral intake (solid, soft or liquid diet) and (4) (naso)gastric-tube dependency. Speech was assessed as the ability to speak using a tracheoesophageal puncture (TEP) prosthesis.

### Statistical Analysis

Pooled outcome rates were calculated using the weighted total number of cases across the included studies, divided by the total number of patients in the respective studies. A random-effects meta-analysis was performed for all outcomes, since at least some heterogeneity was present in most outcomes. All statistical analyses were performed using R software for statistical computing (version 4.0.3). Two-sided *p*-values < 0.05 were considered statistically significant.

## 3. Results

In total, 23 eligible studies for partial hypopharyngeal defects following TLE were selected for inclusion. The included studies reported data on complications and functional outcomes after PMMC flap (n = 13) [9,11,12,13,14,15,16,17,18,19,20,21,22], pectoralis major myofascial flap (PMMF flap, n = 2) [5,9], RFFF (n = 5) [23,24,25,26] and ALTFF (n = 7) [22,26,27,28,29,30,31,32] reconstructions. The numbers of patients included in each reconstructive group were the following: 394 (PMMC flap), 57 (PMMF flap), 160 (RFFF) and 183 (ALTFF), with a total of 794 patients. An overview of all included studies with primary and secondary outcomes is shown in Table 1 and Table 2, respectively. Only 6 out 23 studies were considered to be of good quality using the NIH quality assessment tool (Table A1). Forest plots for fistulas, strictures and oral intake are shown in Figure 3, Figure 4 and Figure 5, respectively. Funnel plots of the primary outcome measures are shown in Figure A1 (online supplement), showing considerable asymmetry for studies reporting on fistula and stricture rates, indicating a high heterogeneity and the risk of publication bias.

### 3.1. Fistulas

Only 8 out of 23 studies described the management of fistulas in a qualitative manner [7,11,13,14,21,22,31]. Therefore, the occurrence of persistent fistulas requiring surgical treatment could not be determined for most studies. Fistulas—regardless of their management—were most frequently reported following PMMC flaps (n = 145/360, 40%), followed by RFFFs (n = 32/160, 20%), ALTFFs (n = 14/142, 10%) and PMMF flaps (n = 4/57, 7%; Table 1). There was considerable heterogeneity between studies, with an I2 value of 80% *p* < 0.001).

The meta-analysis showed a significant difference in fistula incidence (any management) between the groups (*p* < 0.001) and was highest in PMMC flaps with 34% (95% CI 23–47%), followed by RFFF (17%; 95% CI 9–31%), ALTFF (10%; 95% CI 6–17%) and PMMF flap (7%; 95% CI 3–17%) reconstructions.

Rates for surgically treated fistulas were 13% (95% CI 6–28%) for ALTFF, 5% (95% CI 2–13%) for PMMC and 2% (95% CI 0–11%) for PMMF reconstructions, respectively. The pooled incidence of surgically treated fistulas in RFFFs could not be determined due to a lack of studies. No meta-analysis was performed for fistulas requiring surgical management due to a lack of studies.

Conservatively treatable fistulas occurred in 3 of 57 PMMF flaps (5%) followed by 87 of 245 PMMC flaps (36%) in studies that provided details about fistula management. The pooled incidence of conservatively treatable fistulas was significantly different (*p* = 0.02) between PMMC (29%, 95% CI 16–47%) and PMMF flap (5%, 95% CI 1–19%) reconstructions.

### 3.2. Strictures

Stricture occurrence was most frequently reported in PMMC flap reconstructions (n = 33/232, 14%), followed by RFFF (n = 9/76, 12%), ALTFF (n = 6/113, 5%) and PMMF flap (n = 3/57, 5%) reconstructions. The meta-analysis did not identify a significant difference between the groups, with a pooled incidence of 9% in RFFFs (95% CI 1–58%), followed by PMMC flaps (9%; 95% CI 4–20%), PMMF flaps (5%, 95% CI 1–20%) and ALTFFs (5%, 95 CI 1–17%).

### 3.3. Flap Failure

Overall, flap failure rates were very low. Flap failure was reported in 3% of ALTFF cases (n = 3/87), followed by 1% in PMMC flaps (n = 3/260) and RFFFs (n = 1/92), respectively. There were no reported flap failures in PMMF flap reconstructions (n = 0/57). No statistically significant differences were found between the pooled flap failure rates of different reconstructive methods. No rates of partial flap failure or partial necrosis were reported in any reconstructive group.

### 3.4. Functional Outcomes

#### 3.4.1. Swallowing and Dietary Intake

Oral intake (either solid/soft/liquid) was achieved in 96% (95% CI 85–99%) of patients following any reconstruction. No statistically significant difference was observed in oral intake rates between flap types, with 78% (95% CI 59–90%) in ALTFF reconstructions, 93% in PMMC flaps (95% CI 75–98%) and 100% in both PMMF (95% CI 0–100%) and RFFF (95% CI 0–100%) reconstructions, respectively. Most studies did not discern between a solid/soft and liquid diet.

The pooled incidence of (naso)gastric-tube dependency was highest in patients with an ALTFF (20%, 95% CI 10–36%), compared to PMMC (5%, 95% CI 1–30%), PMMF (0%, 95% CI 0–100%) and RFFF reconstructions (0%, 95% CI 0–100%), although this difference was not statistically significant.

#### 3.4.2. Speech

TEP speech was achieved in 59% (95 of 160) of patients following PMMC flap reconstruction, with a pooled effect estimate of 39% (95% CI 12–76%). In comparison, TEP speech was achieved in 89% of patients (25 of 28) that received a RFFF reconstruction, with a pooled effect estimate of 96% (95% CI 79–100%). Only one study reported speech outcomes following ALTFF (20/45, 44%) and PMMF (7/13, 54%) reconstructions, respectively. No reasons were reported for not achieving TEP speech in studies on PMMC flap reconstructions that had low rates of 20% and in which TEP was only tried in a small selection of patients. Due to the sparsity of data, no statistical analyses could be performed.

## 4. Discussion

Over time, various reports on the techniques used to successfully reconstruct partial hypopharyngeal defects after TLE have been published. While the debate on the use of pectoralis major or free flaps in head and neck reconstruction has been ongoing in recent decades, many studies have compared their outcomes across the vast array of head and neck defects in which these reconstructions are being employed. Few studies have directly compared different flap types to assess their superiority in terms of morbidity, functionality and patient-reported quality of life. Consequently, considering the lack of international consensus and sparse comparative data, the choice of reconstruction nowadays is predominantly based on institutional preferences. Nonetheless, many authors reporting on the free-flap reconstruction of partial hypopharyngeal defects advocate for the use of free flaps over pectoralis major flaps based on their respective complication rates [1,26,28,30,33]. However, the results of this meta-analysis do not fully corroborate these recommendations. While fistulas are more common in PMMC flaps than in PMMF and free-flap reconstructions, no significant differences exist in stricture and flap failure rates. Remarkably, reported oral intake and (naso)gastric-tube dependency rates were substantially worse for ALTFF reconstructions. In contrast, PMMF flaps yield comparable results compared to free flaps.

### 4.1. Fistulas

Fistulas are a common and potentially lethal complication in hypopharyngeal reconstruction, as they can lead to the development of secondary issues such as dysphagia, aspiration, flap failure, the delay of subsequent therapy or even vascular blow-out. Moreover, fistulas inherently result in longer hospital stays, a decrease in quality of life, and increased healthcare costs [23,26]. Therefore, limiting the risk of fistula formation is crucial.

Various authors advocate for the use of free flaps because of their lower risk of fistula formation compared to PMMC flaps. In this review, the pooled incidence of unspecified fistulas in PMMC flaps (34%) was observed to be significantly higher than in other flap types. However, it is crucial to take into account that a higher rate of unspecified fistulas in PMMC flaps does not properly reflect the associated morbidity of a frank fistula requiring surgical closure. This may lead to a misrepresentation of clinical data which hampers a valid comparison between studies.

In contrast, PMMF flaps appear to yield promising outcomes in terms of a similar fistula rate (7%) compared with free-flap reconstructions (RFFF 17%; ALTFF 10%) and a significantly lower rate than in PMMC flap reconstructions (34%) [5,9]. The single-stage technique and idea behind a skin-lined or solely mucosal neopharyngeal wall was first proposed by Robertson and Robinson in 1985, who reported successful results in a series of seven patients, of whom only one developed a fistula that could be managed conservatively [7]. Moreover, the harvesting of a PMMF flap in females using the inframammary fold, described by Shindo, provides an excellent alternative to a PMMC flap, while avoiding unnecessary bulk and decreased pliability [34]. Nonetheless, while preventing the common disadvantages of the traditional PMMC, including tissue bulk, poor pliability, intraluminal hair growth and the poor vascularization of the skin island (specifically in obese patients and women), the PMMF flap technique has been reported very scarcely in the literature [5,9].

The variability in the definition of fistulas in the literature complicated the comparison and pooling of fistula rates in the present review. Salivary leakage may spontaneously close over time with conservative treatment, obviating the need for surgical intervention. The majority of fistula cases can be treated conservatively: in studies that did specify the management of fistulas, only 5% of fistulas in PMMC flap reconstructions had to be treated surgically. For frank fistulas, surgical intervention is imperative, since these do not close due to the (neo)epithelization of the fistula tract. For subsequent studies, we suggest defining any anomalous connection either as a fistula, if surgical closure is required, or suture line leakage, if the tract heals by itself using conservative therapy.

### 4.2. Strictures

Stricture rates did not differ significantly between PMMC flap (9%), RFFF (9%), PMMF flap (5%) and ALTFF (5%) reconstructions. In the previous literature, flap bulkiness appears to increase the risk of stenosis in the reconstruction of circumferential hypopharyngeal defects [35]. Using a more pliable flap, a larger neo-hypopharyngeal lumen can be created with a decreased risk of strictures. This suggests that in the context of partial hypopharyngeal defect reconstruction, stricture rates may be reduced by using either a PMMF flap or a thin free flap [5,9].

### 4.3. Flap Failure

Both regionally pedicled and free-flap reconstructions appear to be reliable methods for reconstructing partial hypopharyngeal defects with low flap failure rates (<4%). Partial flap necrosis was not reported in the included studies, and while scarcely mentioned in the literature, higher rates have been reported for PMMC flaps (4–29%) versus lower rates in free flaps [36,37]. Nonetheless, the poor vascularization of the skin island at its most distal border in the absence of documented necrosis may explain the higher fistulas rates in PMMC flap reconstructions. Specifically in the salvage setting after prior (chemo)radiotherapy, this may increase the risk of fistula formation.

### 4.4. Functionality and Quality of Life

Oral intake, swallowing and speech: Pooled data from this review show significantly worse outcomes for ALTFF reconstructions in terms of oral intake. Although no statistical significance was observed, the pooled incidence rate of (naso)gastric-tube dependency in ALTFF reconstructions (20%) was substantially higher compared with PMMC (5%), PMMF (0%) and RFFF (0%) reconstructions. Notably, the low rate of oral intake in ALTFF reconstruction was mostly influenced by the low oral intake (52%) and high (naso)gastric-tube dependency (48%) rates in the study by Harris et al. [22]. Similarly, this was also the case for their reported rates of PMMC flaps, where 50% of patients achieved an oral diet, whereas 50% of patients were tube-dependent. These rates do indicate problems specific to ALTFF or PMMC reconstructions, but may have been the result of differences in the flap inset or the short length of follow-up. It is also noteworthy that in the study by Harris et al., the surgeons opted to reconstruct all pharyngeal defects instead of performing primary closures, even though all defects could have been closed primarily [22].

Taking the sparsity of functional data in free-flap reconstructions into account, no definitive conclusion regarding the superiority of any flap in terms of oral intake can be made. In addition, no significant differences between primary closure, PMMC, ALTFF and RFFF reconstructions have been identified in prior studies [22,30,38]. As was evident from the current review, data on TEP speech in patients following partial hypopharyngeal defect reconstruction are minimal and mostly outdated.

### 4.5. Comparison with Prior Meta-Analyses

The current study with 794 patients is the largest systematic review and meta-analysis on the reconstruction of partial hypopharyngeal defects to date. In addition, no prior review has included PMMF reconstructions. A prior meta-analysis by Koh et al. only included 164 patients with free-flap reconstructions (ALTFF 117 and RFFF 37) [39]. In their analysis, fistula rates were significantly higher following RFFF reconstructions (RR 2.88, 95% CI = 1.33–6.22), whereas no differences were observed for stricture and oral intake rates [39]. A recent network meta-analysis by Costantino et al. examined PMMC (n = 62), ALTFF (n = 104) and RFFF (n = 66) reconstructions, and only included 232 patients [40]. In contrast, the authors advocated for the use of RFFF reconstructions, as their results showed that RFFFs carry the lowest absolute risk for fistulas, strictures and feeding-tube dependency [40].

The large difference in the number of patients compared to the current meta-analysis illustrates the increased power of our meta-analysis. Moreover, the contrasting results and recommendations of prior meta-analyses highlight the ongoing discussion on optimal flap selection for partial hypopharyngeal defect reconstructions.

### 4.6. Cost-Effectiveness

The assessment of the cost-effectiveness of PMMC, PMMF and free-flap reconstructions is challenging, considering the vast array of indications these flaps are used for in oncological head and neck reconstructions [37]. No studies have specifically addressed cost-effectiveness following partial hypopharyngeal defect reconstruction. In contrast, several studies have investigated the cost-effectiveness of onlay flaps for the reinforcement of the hypopharyngeal suture line following salvage total laryngectomy [41,42]. The trade-offs between the use of regional or free flaps have been more extensively studied in oropharyngeal cancer reconstructions [43,44]. Although surgical costs are higher for free-flap reconstructions in general, the overall costs are reported to be less than that of PMMC flap reconstructions due to a significantly shorter length of stay and a lower rate of complications [43,44]. Future studies should focus specifically on hypopharyngeal reconstructions and elucidate which flap types are most cost-effective.

### 4.7. Global Reconstruction Trends: Pectoralis Major Flaps

It is important to realize that complication rates for pectoralis major flap reconstructions have been reported to have improved over the years since the 2000s in both developing and developed countries [45]. This may be related to new anatomical insights and the various technical modifications reported over the years. An overview of these modifications for PMMC reconstructions is shown in Table 3.

Pectoralis major flaps remain of great use in developing countries with limited (microsurgical) resources, and the popularity and initial indications have remained the same [46]. This may also explain why the included PMMC flap studies in this review were predominantly from institutions in developing countries. Interestingly, reports of PMMF flaps were from European countries with high levels of expertise in free-flap reconstructions (Italy and The Netherlands) [5,9].

**Table 3 cancers-16-01804-t003:** Summary of reported modifications in pectoralis major flap reconstructions in order to increase flap viability and decrease the risk of recipient-site morbidity.

Modifications
Avoidance of small skin paddle dimensions (survival is more doubtful with decreasing size) [47]; Skin paddle caudomedially to the nipple [48]; Inclusion of the third or fourth intercostal artery perforator from the internal mammary artery [47,48] and/or preservation of the lateral thoracic artery [49]; Use of a salivary bypass tube may decrease fistula incidence [50].

### 4.8. Practical Recommendations

In the process of determining the most suitable type of flap for each individual patient, it is important to consider the advantages and disadvantages per reconstructive technique, as shown in Table 4. Whenever a regional flap or free flap is used as first-line reconstruction, it is imperative to recognize that any subsequent procedure (e.g., following complications) will be more challenging. Therefore, it is critical to select a flap that carries the lowest chance of re-intervention. Based on the results of the present review, free fasciocutaneous and regional PMMF flaps appear to be the most suitable options to reconstruct a partial hypopharyngeal defect.

### 4.9. Limitations and Future Research

This systematic review has several limitations. First, the heterogeneity of the included studies, including the large discrepancies in the reported outcome definitions, statistical analyses and quality of evidence, made the comparison between different studies challenging. The funnel plot asymmetry likely reflects the high heterogeneity between studies, which might also be (partially) caused by selective outcome reporting. Second, only two studies regarding PMMF flap reconstructions were available in the literature. Although the reported outcomes of the PMMF flap appear promising, further research is warranted to corroborate these results. Third, various authors combined head and neck reconstructive techniques without a specific focus on hypopharyngeal reconstruction, making data extraction from these studies impossible. Other studies focusing on hypopharyngeal reconstruction provided inadequate data, as the majority did not report the management of partial and circumferential defects separately, while both require a different reconstructive approach. Fourth, we could not control for confounding variables (e.g., prior radiotherapy and the extent of the defect) that could have influenced flap choice and reconstructive outcomes. Finally, very few studies reported long-term results, and there is an evident lack of data on health-related quality of life using patient-reported outcome measures in both the short and long term [2].

Standardized outcome parameters in hypopharyngeal reconstruction are required to limit heterogeneity and increase the comparability among (inter)national studies, using a core outcome set. To improve the quality of life of patients undergoing hypopharyngeal reconstruction, future comparative (multicenter) studies should be performed that focus solely on this select group of patients, using valid and reliable patient-reported outcome measures.

## 5. Conclusions

This is the largest systematic review and meta-analysis on partial hypopharyngeal defect reconstruction to date, which identified a higher rate of fistulas following pectoralis major myocutaneous flap reconstructions compared to other flaps, while no other significant differences in terms of postoperative complications were identified. Pectoralis major myofascial flaps yield promising results, with non-inferior postoperative complication rates compared to free-flap reconstructions, warranting further investigation. In order to limit the rate of morbidity following partial hypopharyngeal defect reconstruction, pectoralis myocutaneous flaps should not be the preferred method of reconstruction for most patients. Both free flaps and pectoralis myofascial flaps carry the least morbidity and appear to be the most reliable methods of reconstruction. Any reconstruction should be tailored to each patient depending on local anatomy, co-morbidity and patient preferences, and the final choice should be made through shared decision making after the patient has been informed about the advantages and disadvantages of all reconstruction options.

## Figures and Tables

**Figure 1 cancers-16-01804-f001:**
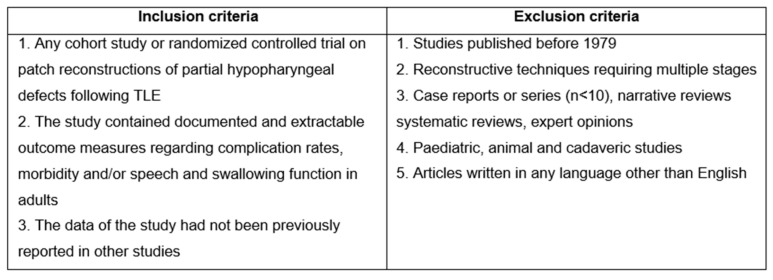
Inclusion and exclusion criteria used for study selection.

**Figure 2 cancers-16-01804-f002:**
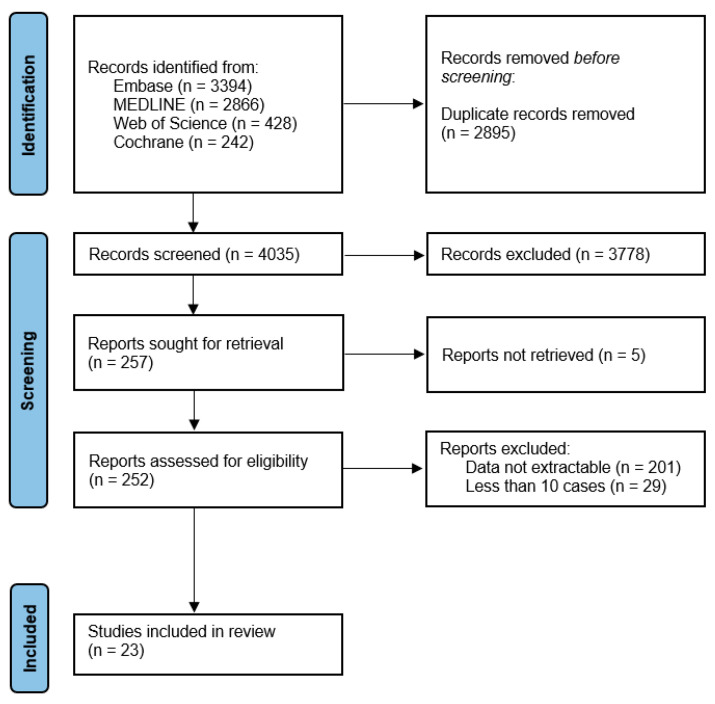
Flow diagram of the inclusion process according to the PRISMA 2020 guidelines.

**Figure 3 cancers-16-01804-f003:**
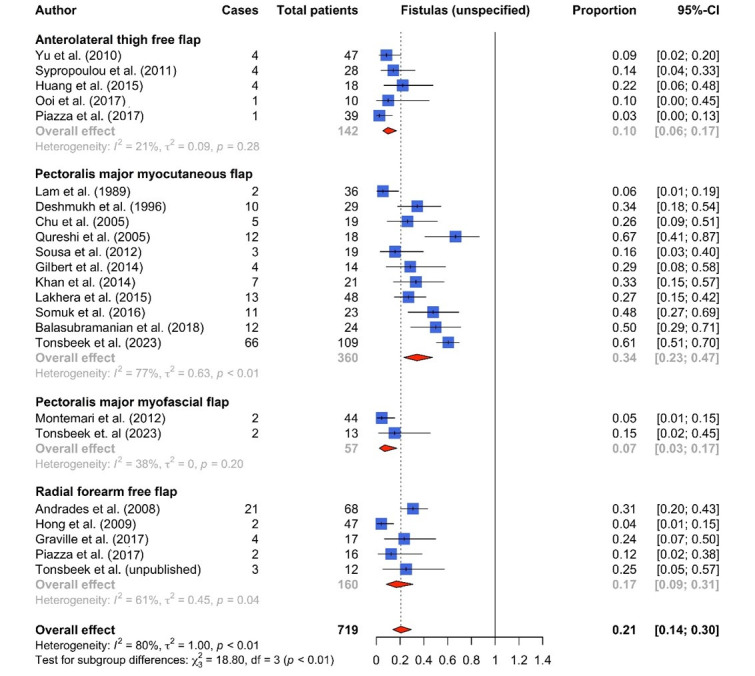
Forest plot of unspecified fistula rates by flap type [5,9,11,12,14,15,16,17,18,19,20,21,23,24,25,26,28,29,31,32].

**Figure 4 cancers-16-01804-f004:**
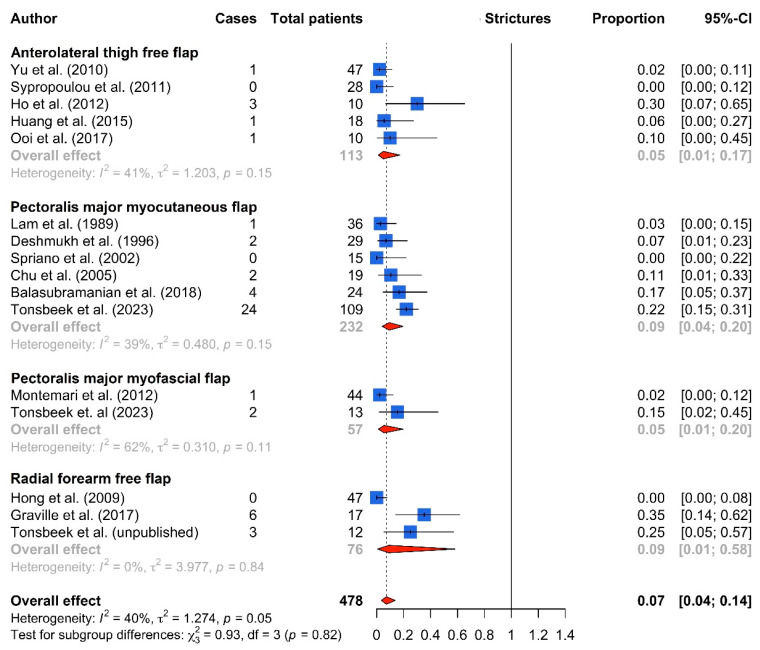
Forest plot of stricture rates by flap type [5,9,11,12,13,14,21,24,25,27,28,29,31,32].

**Figure 5 cancers-16-01804-f005:**
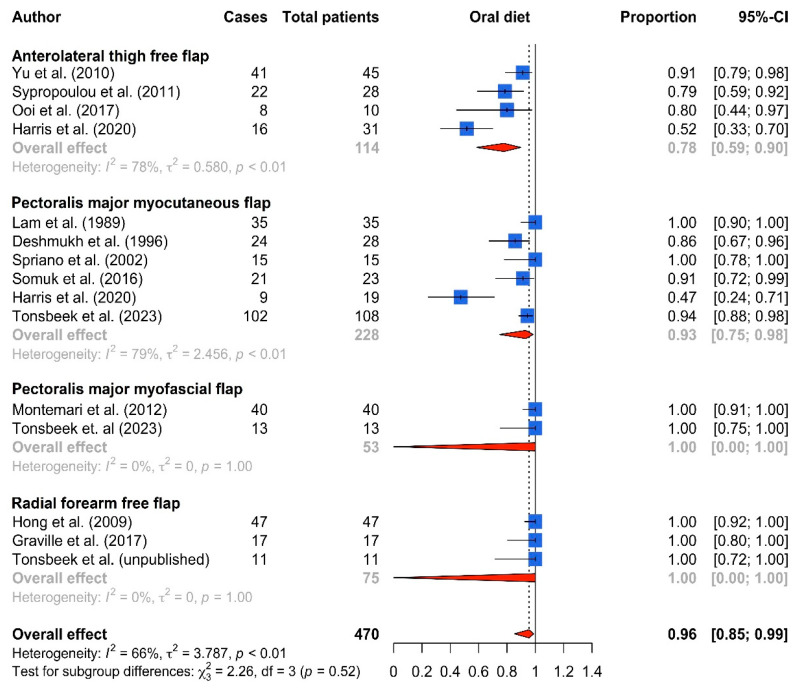
Forest plot of oral intake rates (either solid, soft or liquid diet) by flap type [5,9,11,12,13,20,22,24,25,29,31,32].

**Table 1 cancers-16-01804-t001:** Study characteristics and reported outcomes on fistulas and strictures of the included studies for PMMC, PMMF, RFFF and ALTFF reconstructions.

Study	n	Fistulas ^a^ (n)	Strictures (n)
**Pectoralis major myocutaneous flaps**			
Lam et al. (1989) [11]	36	2	1
Deshmukh et al. (1996) [12]	29	10	2
Spriano et al. (2002) [13]	15	-	0
Chu et al. (2005) [14]	19	5	2
Qureshi et al. (2005) [15]	18	12	-
Sousa et al. (2012) [16]	19	3	-
Gilbert et al. (2014) [17]	14	4	-
Khan et al. (2014) [18]	21	7	-
Lakhera et al. (2015) [19]	48	13	-
Somuk et al. (2016) [20]	23	11	-
Balasubramanian et al. (2018) [21]	24	12	4
Tonsbeek et al. (2023) [9]	109	66	24
**Pectoralis major myofascial flaps**			
Montemari et al. (2012) [5]	44	2	1
Tonsbeek et al. (2023) [9]	13	2	2
**Radial free forearm flaps**			
Andrades et al. (2008) [23]	68	21	-
Hong et al. (2009) [24]	47	2	0
Graville et al. (2017) [25]	17	4	6
Piazza et al. (2017) [26]	16	2	-
Tonsbeek et. al.(unpublished)	12	3	3
**Anterolateral thigh free flaps**			
Yu et al. (2010) [32]	47	4	1
Spyropoulou et al. (2011) [31]	28	4	0
Ho et al. (2012) [27]	10	-	3
Huang et al. (2015) [28]	18	4	1
Ooi et al. (2017) [29]	10	1	1
Piazza et al. (2017) [26]	39	1	-

**^a^** Includes both reported surgically and conservatively treated fistulas, because in many studies the difference was not reported.

**Table 2 cancers-16-01804-t002:** Study characteristics and reported outcomes of the included studies for PMMC, PMMF, RFFF and ALTFF reconstructions on flap failure and functionality (speech and oral intake).

Study	n	Flap Failure	TEP Speech ^a^	Diet		
				Solid/Soft	Liquid	Tube Dependent
**Pectoralis major myocutaneous flaps**						
Lam et al. (1989) [11]	35	1	20%	94%	6%	0%
Deshmukh et al. (1996) [12]	28	0	-	85%		-
Spriano et al. (2002) [13]	15	0	20%	87%	13%	0%
Lakhera et al. (2015) [19]	48	0	-	-	-	-
Somuk et al. (2016) [20]	23	0	-	22%	69%	9%
Balasubramanian et al. (2018) [21]	24	-	-	79%	-	-
Harris et al. (2020) [22]	19	-	-	50%		50%
Tonsbeek et al. (2023) [9]	108	2	80%	94%		6%
**Pectoralis major myofascial flaps**						
Montemari et al. (2012) [5]	40	0	-	100%	0%	0%
Tonsbeek et al. (2023) [9]	13	0	54%	100%		0%
**Radial free forearm flaps**						
Hong et al. (2009) [24]	47	0	-	100%	0%	0%
Graville et al. (2017) [25]	17	0	82%	100%		0%
Piazza et al. (2017) [26]	16	1	-	-	-	-
Tonsbeek et al. (unpublished)	11	0	100%	100%		0%
**Anterolateral thigh free flaps**						
Yu et al. (2010) [32]	45	-	44%	91%		9%
Spyropoulou et al. (2011) [31]	28	1	-	75%	4%	21%
Ho et al. (2012) [27]	10	0	-	-	-	-
Huang et al. (2015) [28]	18	-	-	-	-	11%
Ooi et al. (2017) [29]	10	1	-	70%	10%	20%
Piazza et al. (2017) [26]	39	1	-	-	-	-
Harris et al. (2020) [22]	31	-	-	52%		48%

^a^ Percentage of patients that achieved TEP speech. Studies without extractable functional outcomes or data on flap failure were not included in the table.

**Table 4 cancers-16-01804-t004:** Summary of reported advantages and disadvantages between pectoralis major and free flaps.

	PMMC/PMMF	Free Flap (ALTFF/RFFF)
**Surgical aspects**	PRO: relative ease of harvesting, reliable blood supply, large tissue volume useful for coverage, short surgical time required for reconstruction and suitable for patients with severe comorbidities or a vessel-depleted neck. CON: less pliability (predominantly PMMC) due to tissue bulk, more complicated flap inset, limited cranial reach due to pedicle and risk of an unviable skin island.	PRO: good pliability due to minimal tissue bulk, a plannable flap size to match the defect and better postoperative functionality. CON: more challenging to harvest, and longer surgery time and less suitable for patients with severe comorbidities or a vessel-depleted neck.
**Recipient-site morbidity**	Fistula, conservative PMMC ↑ PMMF = Fistula, surgical = Stricture =	Fistula, conservative **↓** Fistula, surgical = Stricture =
**Donor-site morbidity**	Uncommon (±4% of cases) [51] Most frequent: wound infection (4–11%) [50]	Infrequent (<5% of cases with wound dehiscence, seroma, hematoma or infection) [52,53] Most frequent: paresthesia or dysesthesia (±25%) [52,53]
**Functionality** Oral intake (any oral diet) Tube dependency Speech	Oral intake = Tube dependency = Insufficient data	Oral intake ALTFF ↓ RFFF = Tube dependency ALTFF ↑, RFFF = Insufficient data
**Cost-effectiveness**	Unclear, no specific data are available on cost-effectiveness in partial hypopharyngeal defect reconstruction.	

Abbreviations: ALTFF (anterolateral thigh free flap), PMMC (pectoralis major myocutaneous flap), PMMF (pectoralis major myofascial flap) and RFFF (radial forearm free flap). Legend: ↑ = relatively higher rate, ↓ relatively lower rate, = relatively similar rate.

## Data Availability

The original contributions presented in the study are included in the article, further inquiries can be directed to the corresponding author.

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
