# Peer review of "Reconstruction of Partial Hypopharyngeal Defects following Total Laryngectomy: A Systematic Review and Meta-Analysis"

_cancers, 2024, doi:10.3390/cancers16101804_

Round 1

Reviewer 1 Report

Comments and Suggestions for Authors

Line 51-52 “A reconstruction is required if less than ± 3 cm of stretched hypopharyngeal mucosa remains.” Please clarify what ± means

Line 228 “Few comparative studies focused specifically  on hypopharyngeal defect reconstruction to investigate which reconstructive method yields superior results” the authors should revise to clarify the meaning

Comments on the Quality of English Language

see comments

Reviewer 2 Report

Comments and Suggestions for Authors

The authors performed meta-analyses to investigate the complication rate and functional result of pedicled or free flap reconstruction for partial hypopharyngeal defect after total laryngectomy. Their largest meta-analysis including 794 patients must have more scientific power than previous ones. They concluded that PMMC flap had a higher rate of salivary leakage than  PMMF, ALTFF, and RFFF in hypopharyngeal reconstruction. While there was no difference about stricture and oral intake in pedicled and free flaps, speech function could not analyzed because of shortage of data. Their observation is giving an impact for head and neck surgeons. Some minor issues should be reconsidered.

Line 298, "Harris et al.22" should be shown with a correct form.

In Table 4, Surgical aspects of Free flap, "OR" needs an explanation.

4.7. Cost-effectiveness section needs more information and assessment about cost-effectiveness of PMMC, PMMF, and free flap reconstruction.

Reviewer 3 Report

Comments and Suggestions for Authors

The study's strength lies in its extensive review process, though the significant heterogeneity noted among the included studies might dilute the strength of the conclusions drawn. The tables and figures are detailed and enhance understanding, but the high variability in study designs included could confuse less expert readers. The findings are highly relevant clinically and are likely to influence both surgical practice and guidelines. However, the paper could benefit from a clearer summary of practical recommendations based on flap types. The study calls for more standardized research in this area, which is crucial. It suggests a move towards more homogenous study designs to better compare outcomes across studies. This paper is a commendable effort to synthesize available data on a complex topic, providing insights that are likely to benefit surgical practice. While the heterogeneity of the included studies poses a challenge, the careful analysis and comprehensive review process add significant value to the field of reconstructive surgery. The call for more uniform study designs is pertinent and should guide future research efforts in this area.
